# Factors Associated with the Use of Pasteurized Donor Milk for Healthy Newborns: Experience from the First Human Milk Bank in Vietnam

**DOI:** 10.3390/nu13041151

**Published:** 2021-03-31

**Authors:** Hoang Thi Tran, Tuan T Nguyen, Hoang Thi Nam Giang, Le Thi Huynh, Debbie Barnett, Roger Mathisen, John C. S. Murray

**Affiliations:** 1Neonatal Unit and Human Milk Bank, Da Nang Hospital for Women and Children, Da Nang 50506, Vietnam; bongdiendien26@gmail.com; 2Department of Pediatrics, School of Medicine and Pharmacy, The University of Da Nang, Da Nang 50206, Vietnam; 3Alive & Thrive Southeast Asia, FHI 360, Hanoi 11022, Vietnam; tnguyen@fhi360.org (T.T.N.); rmathisen@fhi360.org (R.M.); 4International Cooperation and Scientific Research Office, School of Medicine and Pharmacy, The University of Da Nang, Da Nang 50206, Vietnam; htngiang@ud.edu.vn; 5Milk Bank Scotland, Queen Elizabeth University Hospital, Glasgow G51 4TF, UK; Debbie.Barnett@ggc.scot.nhs.uk; 6International consultant, Iowa City, IA 52240, USA; jcsmurray@hotmail.com

**Keywords:** early essential newborn care, human milk bank, pasteurized donor milk, perinatal, prelacteal feeding, term newborn

## Abstract

Background: Pasteurized donor milk (PDM) is typically prescribed to preterm or low birthweight newborns when their mother’s own milk is unavailable. In surplus, PDM is prescribed to meet the nutritional needs of healthy newborns in the first few days of life. However, its overuse can undermine efforts to promote and support breastfeeding, waste resources, and reduce the availability of PDM for at-risk newborns. We conducted this study to examine factors associated with the prescription and prolonged use (>48 h) of privately purchased PDM to healthy newborns. Methods: Prospective observational study of 2440 mothers of healthy, term, and normal birthweight newborns born at Da Nang Hospital for Women and Children between April and August 2019. In addition to the descriptive analysis, we performed multiple logistic regressions to examine factors associated with the prescription of PDM (*n* = 2440) and prolonged PDM use among those who used PDM (*n* = 566). Results: Twenty-three percent (566/2440) of healthy, term, and birthweight ≥2500 g newborns received PDM and were included in the study. The prevalence of PDM use was higher for cesarean births (OR: 2.05; 95% CI: 1.66, 2.55) and among male newborns (OR: 1.33; 95% CI: 1.09, 1.62), but lower for farmers or workers (vs. other jobs; OR: 0.71; 95% CI: 0.54, 0.93), family income <10 million VND (vs. ≥10 million VND; OR: 0.67; 95% CI: 0.55, 0.82), and duration of skin-to-skin ≥90 min (vs. <90 min; OR: 0.54; 95% CI: 0.39, 0.76). Prolonged PDM use (12.4% of 566 newborns who used PDM) was associated with the mother having a higher socioeconomic status job (professional, small trader or homemaker; OR: 4.00; 95% CI: 1.39, 12.5), being a first-time mother (OR: 3.39; 95% CI: 1.92, 6.01) or having a cesarean birth (OR: 2.09; 95% CI: 1.02, 4.28). Conclusions: The prescription and prolonged use of privately purchased PDM was associated with non-medical factors unrelated to the ability to breastfeed effectively. The findings suggest the need for improved breastfeeding communication, counseling and support skills for health staff, development, and application of strict criteria on PDM use for healthy newborns and better routine monitoring of PDM use over time.

## 1. Introduction

Newborn deaths accounted for more than 47% of under-5 mortality globally in 2018, with 98% of these deaths occurring in low- and middle-income countries [1,2]. In Vietnam, neonatal mortality declined from 24 to 11 per 1000 live births between 1990 and 2018 and now represents around 52% of under-5 deaths [1]. Common causes of neonatal deaths include complications of prematurity, infections, birth asphyxia, and congenital malformations [3]. Early and exclusive breastfeeding is the most effective intervention to reduce preventable neonatal mortality and morbidity. Breastfeeding in the first hour of life is estimated to prevent up to 22% of neonatal deaths [4] and to reduce the risk of death due to neonatal infections by 2–12 times [5]. In addition, breastfeeding promote long term infant growth and development [6]. However, not all newborn infants have access to their mother’s own milk in the first hour after birth and in the early newborn period. Common reasons include mother’s absence due to separation, sickness, death or abandonment; inability to breastfeed due to a medical condition or use of prohibited medications; and no lactation capacity including adoptive parents [7]. Wet nursing or informal sharing of other mothers’ milk is sometimes considered an antidote to this problem, but this practice increases the risk of infection transmission [8]. The provision of safe pasteurized donor milk (PDM) is now common in some countries through human milk banks (HMB), and represents a potential solution [7,9].

There are a number of benefits from the use of PDM, including reduction in the risk of severe infections and necrotizing enterocolitis [10] and neonatal sepsis in the first 28 days of life among low birthweight infants [11], and retinopathy of prematurity, bronchopulmonary dysplasia, and duration of ventilator support among preterm infants [12]. Furthermore, breastfeeding rates among infants discharged from neonatal units using PDM are higher than those not using PDM [12,13,14]. Therefore, when quantities of PDM are limited, use by sick or preterm newborns should be prioritized [9].

In surplus, PDM is prescribed for otherwise healthy newborns with risk factors for interruption of breastfeeding [15,16,17]. Although there is no specific guidance on the prescription of PDM to healthy newborns, the most common indications include hypoglycemia and/or hyperbilirubinemia, excessive weight loss, delayed lactogenesis, small for gestational age, and maternal-infant separation [17]. Disparities in accessing PDM according to race/ethnicity, socio-economic status, and access to private health insurance have been noted [17]. Feeding of PDM to healthy newborns can occur within the network of an HMB. For example, in 2017, a study on the HMB network in the Northeastern United States showed that 32% (23 out of 71) of hospitals provided PDM to healthy newborns [15]. This study showed that the median prevalence of exclusive breastfeeding at discharge was higher in hospitals using PDM for healthy newborns (median: 77%; range: 38–90% vs. median: 56%; range: 7–92% [15].

There is very limited information about the use of PDM for healthy newborns, especially among low- and middle-income countries [17]. A recent review on the use of PDM in populations other than preterm infants identified only 26 studies from the United States (20 studies), Denmark (two studies), South Africa, Poland, and Canada. Of the 26 studies, 14 studies focused on healthy infants born >35 weeks and were conducted in the last six years [17]. Data from the first HMB in Vietnam operating at Da Nang Hospital for Women and Children (DNHWC) in Vietnam, a lower-middle-income country, can help fill this literature gap. Established in February 2017, the HMB has expanded from sick, preterm or low birthweight newborns to include healthy newborn infants who are unable to breastfeed (starting June 2017) [18]. In the first four years of operation, the HMB distributed PDM to 16,235 newborns, with two thirds of the recipients being healthy newborns in postnatal wards [19]. We conducted this study to examine the factors associated with the prescription and prolonged use (>48 h) of privately purchased PDM to healthy newborns. The study findings could be used to inform the development of strategies to improve breastfeeding practices, including ensuring that PDM is not used unnecessarily for healthy newborns.

## 2. Methods

### 2.1. Study Setting

The DNHWC is a tertiary hospital for obstetrics, gynecology, and pediatrics, which has 1200 beds and serves over three provinces with a population of 4.4 million in 2019–2020. More than 15,000 births occur annually. As a referral hospital for the central region of Vietnam, approximately 30% of the hospital admissions are women with high-risk pregnancies and sick children. The implementation of Ten Steps to Successful Breastfeeding and continuing hospital quality improvement activities have contributed to the increased prevalence of early initiation and exclusive breastfeeding for both vaginal and cesarean section births [20,21]. Early essential newborn care, including delayed cord clamping, immediate and prolonged skin-to-skin contact between the mother and newborn for at least 90 min, and completion of the first breastfeed before separation, began in 2014 for both vaginal and cesarean section births [22]. Newborns are kept with mothers under clinical observation for 2 h (vaginal births) or 6 h (cesarean births) and then transferred to postnatal wards if stable [22]. Infant formula is not provided in the hospital, per hospital guidelines [22]. At DNHWC, except for limited cases with medical reasons, infants and mothers stay together (same bed, same room) and have the opportunity to breastfeed on demand. Mothers are counseled to perform hand breastmilk expression or to use their own breast pumps or the hospital pumps at the HMB when needed.

The HMB at DNHWC was established in February 2017 with financial and technical support from the Vietnam Ministry of Health, Da Nang Department of Health, DNHWC, Alive & Thrive, PATH, and international experts [18]. The HMB began providing PDM to all newborns requiring breastmilk from June 2017. Potential donors are recruited from within the hospital or from the community and undergo systematic screening and education prior to approval [18]. Donors volunteer and do not receive any money for the donation. Although the HMB is a non-profit project of the hospital, PDM is purchased by families at a cost of 130,000 VND (USD ~6) per 100 mL, with fees used to cover costs of microbiology tests for milk samples before and after pasteurization, utilities, equipment maintenance, equipment sterilization, electricity, and water [18]. The majority of families offered PDM can pay the fees; DNHWC provides free PDM for families who are poor. Decisions about using PDM are made collaboratively by mothers and clinicians based on the newborn’s feeding behavior after breastfeeding attempts and lactation support. The amount of PDM given is based on the newborn’s age. On the first day of life, eight 10-mL containers of PDM are provided to the family, followed by eight 15-mL containers on day 2 and eight 20-mL containers on day 3. All mothers using PDM receive lactation support from midwives in the postnatal wards. If a newborn requires PDM for more than 3 days, lactation staff from the HMB provide additional support for mothers.

### 2.2. Study Procedures and Participants

This study was a prospective observational study with women in postnatal wards. A short, structured questionnaire was developed to obtain information about maternal characteristics, antenatal and perinatal care. Fifteen senior midwives were trained to administer the questionnaire in three postnatal wards with three team leaders for data collection. Between April and August 2019, all mothers in postnatal wards on weekdays who were within 24 h of discharge (women are typically discharged 3–4 days after vaginal births and 5–7 days after cesarean births), with a term (≥37 weeks’ gestation), birthweight ≥2500 g, healthy newborn, and who had received PDM were asked to participate in the study. If they consented, a senior midwife interviewed the mother. The interview was conducted in a quiet patient room as a part of a routine checkup and consultation, and typically lasted for 10–15 min. At the end of each working day, the questionnaires were submitted to the team leaders who reviewed and approved the questionnaires before sending to a staff member to enter the data into a form in Microsoft Access 2013.

Since the original purpose of the study was to examine the characteristics of newborns receiving PDM, we estimated the sample size based on the random sampling equation for one proportion [23]. A minimum sample size of 600 newborns using PDM was calculated based on a prevalence of 50% to be conservative (i.e., largest sample size), alpha of 0.05, absolute precision (d) of 4% [23]. We estimated that 25% of healthy newborns at DNHWC are prescribed PDM and therefore 2400 healthy mothers and newborns would need to be screened to obtain the required sample size. The secondary purpose of examining factors associated with the prolonged use of PDM arose during the data analysis. The sample size of 566, prevalence of prolonged use of PDM of 12.4%, and alpha of 0.05 correspond to an absolute precision of 2.7% [23].

### 2.3. Study Variables

Dependent variables. The use of PDM was defined by the following questions: “Did you use PDM for your newborn?” If Yes, “How long after birth, was your child given PDM (in hours and minutes)” and “How long after birth did your child stop receiving PDM (in days, hours, and minutes).” The prolonged use of PDM was defined as giving PDM for more than 48 h.

Independent variables. Information collected on the newborn included sex, birth mode, gestational age, birthweight, duration of skin-to-skin contact, and time of first breastfeed. Information collected on the mother included age, number of previous children, education, profession, and medical conditions. Education was classified into four groups: Primary-secondary school, high school, college, and university and postgraduate. Occupation was classified into five groups: Professional, housewife, worker, farmer, and small trader. Family income was classified into four monthly income groups: <5 million VND, 5–10 million VND, 10–20 million VND, and >20 million VND (USD 1 equals 22,900 VND).

### 2.4. Statistical Analysis

Data were entered into a Microsoft Access 2013 form and analyzed using R (Version 3.6.3; https://www.r-project.org; accessed on 15 March 2020). First, we performed descriptive statistics and presented the characteristics of newborns and mothers as the proportion, mean (standard deviation) or median (interquartile range). Second, we performed a multiple logistic regression model (Model 1; *n* = 2440) to examine the associations between the use of PDM and gender (male vs. female), birthweight (>3600 g vs. ≤3600 g), gestational age (continuous), skin-to-skin duration (<90 min vs. ≥90 min), maternal age (continuous), occupation (blue-collar vs. other jobs), education (secondary school education or less vs. a higher education level), income (<10 million VND vs. ≥10 million VND), and mode of birth (cesarean vs. vaginal births), with results presented as odds ratios and 95% confidence intervals (CI). Third, we performed a multiple logistic regression model (Model 2; *n* = 566) to examine the association between the prolonged use of PDM (>48 h) and covariates used in Model 1. The reasons for using a birthweight cut point of 3600 g were, (1) there were only 26 newborns (eight of them used PDM) who weighed >4000 g (heavy for gestational age) and zero newborns weighing >4500 g (exceptionally large baby) in our sample [24], (2) the mean and median birthweights were around 3200 g, and (3) the interquartile range was 3000 and 3500 g. In addition, we visually reviewed the prevalence of prolonged use of PDM by each 100 g of birthweight and found that a birthweight of >3600 g tended to be associated with a higher prevalence of prolonged use of PDM (i.e., change in the slope).

## 3. Results

### 3.1. General Characteristics

Between April and August 2019, there were 4380 healthy newborns born at DNHWC who did not have any medical problems except for mild or moderate jaundice. Of these, 2513 (57.4%) mothers were approached for an interview. We were unable to interview the mothers of the remaining 1867 newborns (42.6% of the total) since they were discharged during the weekend (1270 newborns) or during a weekday when the staff were not available (597 newborns) to conduct the interviews. Of the 2513 mothers approached, 73 (3%) declined to participate. Of the 2440 mothers interviewed, 566 mothers had received PDM during the hospital stay (23.2%) for their newborns and were included in this analysis (Table 1).

Of the 2440 mothers interviewed, the mean age was 29 years; 35.2% were first-time mothers, 41.8% had at least a college education, 45.8% had a professional job; and 32.6% lived in a household with a family monthly income of at least 10 million VND (~USD430), considered higher income in this population (Table 1). The mothers of the newborns who received PDM tended to have a higher socio-economic status compared to those of the newborns who did not receive PDM (Table 1).

### 3.2. Newborn Characteristics

Of the 2440 newborns included, 61.9% were born to mothers who had received prenatal care visits at DNHWC, 59.3% were born by cesarean section, 54.2% were male, and the mean birthweight was 3207 g (Table 2). Most newborns (98.3%) received immediate skin-to-skin contact at birth and 92.4% received prolonged and uninterrupted skin-to-skin contact of at least 90 min. Newborns receiving PDM had a higher prevalence of receiving a perinatal visit at DNHWC, being born by cesarean section, and being male, but a lower prevalence of receiving skin-to-skin contact for at least 90 min (Table 2). Of mothers whose newborns were prescribed PDM, 12 (2.1%) had an underlying medical condition, most frequently hepatitis B, gestational diabetes, and hypertension. The proportion was similar to those whose newborns were not prescribed PDM (Table 2). The proportion of women with reported difficulties breastfeeding was 11.2% among those who prescribed PDM, which was lower than those who were not (17.4%, Table 2).

### 3.3. Factors Associated with the Prescription of PDM

PDM was given to 23.2% of newborns, with 15.9% prescribed before 6 h and 56.4% before 12 h after birth (Figure 1). The prevalence of early PDM use was higher for vaginal (32.5%) than cesarean births (9.5%) (Figure 1). The median duration of PDM use was 24 h (IQR 12–48). The prevalence of PDM use was higher in cesarean births (OR: 2.05; 95% CI: 1.66, 2.55) and among male newborns (OR: 1.33; 95% CI: 1.09, 1.62), but was lower for newborns with mothers being farmers or workers (vs. other jobs; OR: 0.71; 95% CI: 0.54, 0.93), family income <10 million VND (vs. ≥10 million VND; OR: 0.67; 95% CI: 0.55, 0.82), and duration of skin-to-skin ≥90 min (vs. <90 min; OR: 0.54; 95% CI: 0.39, 0.76) (Table 3).

### 3.4. Factors Associated with the Prolonged Use of PDM

The prevalence of prolonged use of PDM (>48 h) was 12.4%. The prevalence of prolonged use of PDM was higher among first-time mothers (OR: 3.39; 95% CI: 1.92, 6.01), cesarean births (OR: 2.09; 95% CI: 1.02, 4.28), and birthweight >3600 g (OR: 3.02; 95% CI: 1.55, 5.89), but lower among farmers or workers than other jobs (OR: 0.25; 95% CI: 0.08, 0.72) (Table 3).

## 4. Discussion

In this study, we found that almost one quarter of healthy, full term, normal birthweight newborns assessed in this study (566 of 2440) were prescribed PDM. Of newborns receiving PDM, 16% were given before 6 h of life and 12.4% used PDM for more than 48 h. Use of PDM was associated with having a cesarean section birth, a male newborn, higher socio-economic status, and with the application of skin-to-skin contact for less than 90 min. Prolonged use of PDM was associated with being a first-time mother and with having a cesarean birth, a higher birthweight newborn, and higher socio-economic status.

A cesarean section was associated with both the use and prolonged use of PDM. Previous studies have found an association between cesarean section and the use of PDM in the United States [15,16,17]. A cesarean section is associated with lower volumes of breastmilk transferred to infants [25], additional birth discomfort, and need for rest [26], all of which may affect the practice of exclusive breastfeeding during the hospital stay and after discharge [25,27]. Prolonged skin-to-skin contact has been demonstrated to be associated with both early and exclusive breastfeeding of hospitalized babies, consistent with the finding that the reduced duration of contact is associated with PDM use [20]. Furthermore, although our study showed a high prevalence of prolonged skin-to-skin contact (92.4%) and a strong protective effect on the use of PDM, the negative impact of cesarean birth remained. Mothers of heavier newborns may experience more discomfort and need more rest due to the prolonged labor, as well as potential concerns that their newborns are less settled and appear hungrier [26]. First-time mothers and family members might have less breastfeeding experience and therefore be less likely to successfully breastfeed compared with more experienced mothers [28].

However, in most cases, the factors associated with the prescription and prolonged use of PDM in this study have no direct link to medical indications that might limit or restrict exclusive breastfeeding. Most perceived barriers after surgery or for perceived maternal pain, discomfort, or capacity, can be overcome with effective counseling and support. Initiation of PDM was associated with a bias among health staff and mothers that selectively allows the use of PDM for some mothers and not for others.

The first possible source of PDM prescription bias may be a perception that if lactation has not begun in the first few hours, that it is not possible. In fact, with the exception of cases of hypoglycemia or dehydration healthy infants have enough reserves to overcome several hours without nutrition providing a wide window of opportunity for initiating feeding [29].

The second possible source of PDM prescription bias may be pressure on health workers from mothers and family members when mothers perceive their own milk as insufficient or unavailable. In the United States, the most common indication for providing PDM was the request of parents or caregivers [16]. The association of PDM use with a higher maternal socio-economic status suggests that the ability of a family to pay for PDM may influence whether or not they receive it. Conversely, affordability may be an obstacle for those with lower incomes, which motivates them to feed the mothers’ own milk. Other studies have shown that when PDM is paid for by private health insurance, this is the case [30,31]. Similarly, disparities in access to PDM according to race/ethnicity, socio-economic status, and access to private health insurance has been reported in the United States [15,16,17].

The third possible source of PDM prescription bias may be the culture around the management of newborns born by cesarean section, and those separate before 90 min of uninterrupted skin-to-skin contact. Factors driving the management of this group of newborns include routines and protocols which treat these newborns as being unable to breastfeed early and effectively, either since the mother or baby are not capable of doing so, or due to a perceived increased risk of complications or illness. The association of prolonged PDM use for male newborns, first-time mothers, and babies over 3600 g may also reflect perceptions about the breastfeeding needs or difficulties likely to be encountered by these groups and therefore a lower threshold for initiating the use of PDM.

Although DNHWC has adopted standard criteria for prescribing PDM by clinicians, including that it cannot be used until after multiple efforts have been made to overcome breastfeeding difficulties [18], these data suggest that better monitoring of the prescription of PDM is necessary, with improved staff coaching in strategies for improving breastfeeding practices including counseling and breastfeeding support. These strategies must address the inaccurate perceptions of risk to the newborn currently held by both mothers and health staff, and how to manage pressure from concerned mothers and families. In addition, stricter criteria for PDM use among healthy newborn infants should be developed, applied, and monitored, in particular for cesarean births, heavier newborns, and for first-time mothers.

For an HMB with a shortage of supply, the use of PDM for non-medical reasons could minimize the availability of PDM for preterm and low birthweight newborns. The HMB at DNHWC has a surplus supply of PDM for its own use and for use by other hospitals. The PDM fee covers the operation of the HMB and is approved by the Da Nang Department of Health [18]. To reduce disparity, the hospital covers the fee for those who cannot afford PDM [18]. Meanwhile, in Vietnam, all hospital fees for children under 6 years old are covered by health insurance [32]. Never-the-less, given the recipients pay for PDM, it remains possible that the hospital, HMB or health workers have an incentive or motivation to selectively prescribe PDM to those who can most afford it. For this reason, ongoing tracking of PDM use by groups who are less likely to be able to afford it, should be included in routine hospital monitoring. Even if the family could pay for the PDM, it could be an unnecessary financial burden for the families.

To our knowledge, this study is among the few studies that have examined the factors associated with the use of pasteurized donor milk for healthy newborns in a lower-middle income country. Several limitations are noted. First, the study sample may not be representative of all women giving birth to heathy newborns during the study period, since weekend discharges and some weekly discharges could not be interviewed due to staffing limitations. However, there are no hospital policies or protocols that would make mothers and healthy newborns discharged at weekends different from those discharged during the week, nor is there evidence that particular groups were systematically discharged during the week without being interviewed. Second, the cross-sectional design means that findings may not represent the quality of hospital services across time that may change with caseloads, case type, availability of staff or staff skills, although no significant variation in these factors are expected. Third, reporting or recall bias by mothers is possible when responding to questions about practices around birth or problems with breastfeeding. All interviewers were trained to ask questions in the same way using standardized structured questionnaires and interviews were all conducted in the early postpartum period to minimize recall bias. The validation of the mothers’ recall of immediate newborn care practices has shown high levels of agreement between observed and reported measures of initiation of skin-to-skin contact, duration of uninterrupted skin-to-skin contact, completion of the first breastfeed during uninterrupted skin-to-skin, and exclusive breastfeeding [33]. Fourth, the use of > 48 h to indicate the prolonged use of PDM was not based on any clinical criteria. Due to the lack of information and agreement relating to this topic [17], we identified this cut-off point based on the distribution of the data and the assumption that all mothers have sufficient breastmilk for their newborns after birth. Fifth, in this study, we did not evaluate the potential impacts to the HMB by fully restricting it to medical use. However, we believed that the HMB will still be functioning since it is the initial design of the HMB to preterm, low birthweight, and sick newborns with the assumption of a low supply of PDM. A costing study is needed to evaluate cost implications of different use scenarios. Finally, we did not evaluate the impact of of PDM use on exclusive breastfeeding, growth or development outcomes because we did not collect these outcomes. Further studies are needed to evaluate the impact of PDM on healthy term newborns.

## 5. Conclusions

The prescription and prolonged use of privately purchased PDM was associated with non-medical factors unrelated to the ability to breastfeed effectively. The findings suggest the need for improved breastfeeding communication, counseling and support skills for health staff, development and application of strict criteria on PDM use for healthy infants, and better routine monitoring of PDM use over time. Further studies to develop regional and international guidelines on the use of PDM for healthy newborns are needed.

## Figures and Tables

**Figure 1 nutrients-13-01151-f001:**
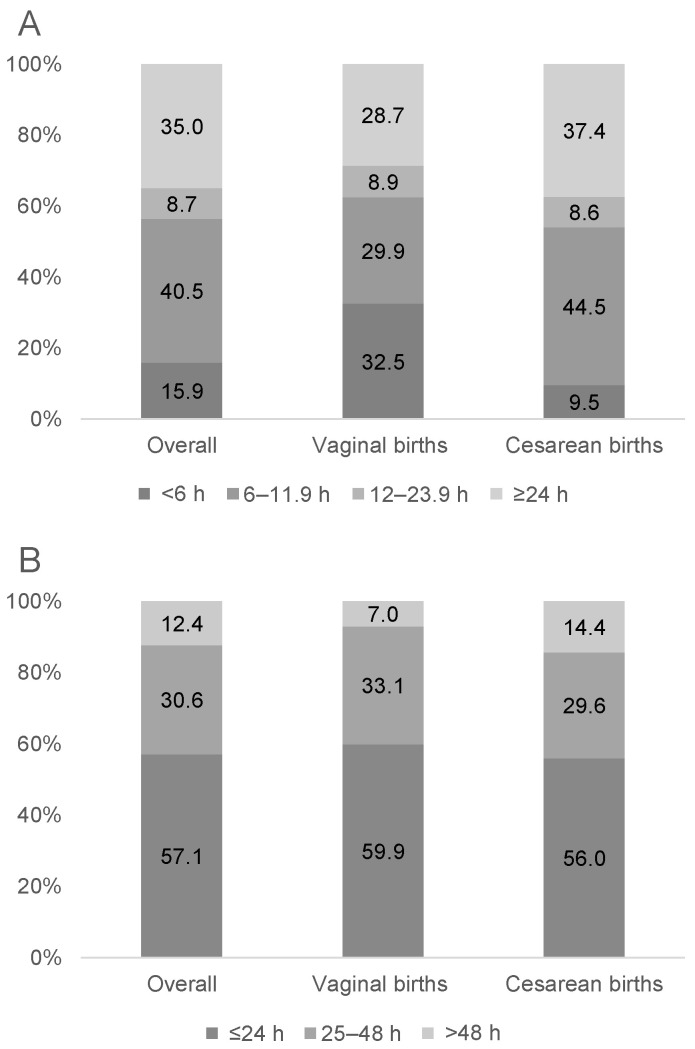
Proportion of newborns prescribed PDM by the time after birth that PDM use began (**A**) and duration of PDM use (**B**) (*n* = 566).

**Table 1 nutrients-13-01151-t001:** Maternal and household characteristics by pasteurized donor milk use.

	Overall (*n =* 2440)	Use PDM (*n =* 566)	Did Not Use PDM (*n =* 1874)
Maternal age (years), mean (SD)	29.0 (5.2)	29.4 (5.8)	28.9 (5.4)
First-time mother	35.2	34.1	35.5
Education:			
Primary and secondary school	7.1	4.9	7.7
High school	50.8	34.5	55.7 *
Diploma, college or higher	41.8	60.6	36.2 *
Occupation:			
Professional jobs (e.g., teacher, doctor, businessperson)	45.8	56.4	42.7 *
Blue-collar jobs (e.g., worker or farmer)	24.4	17.5	26.5 *
Small trader	12.7	12.4	12.9
Homemaker	17.0	13.8	18.0
Family monthly income:			
<5 million VND	7.8	4.8	8.7 *
5–9.9 million VND	59.6	53.5	61.4 *
10 million VND or higher	32.6	41.7	29.9 *

Notes: Data were presented as % or mean (SD) when specified. Abbreviations: *n*: Sample number; PDM: Pasteurized donor milk; SD: Standard deviation; VND: Vietnam Dong, conversion rate of USD 1 for 22,900 VND. * Statistically different from newborns who used PDM, two-sided chi-squared tests for the prevalence and two-sided *t*-test for the mean.

**Table 2 nutrients-13-01151-t002:** Newborn characteristics by pasteurized donor milk use.

	Overall (*n* = 2440)	Used PDM (*n* = 566)	Did Not Use PDM (*n* = 1874)
Prenatal visit at DNHWC	61.9	67.7	60.1 *
Cesarean birth	59.3	72.3	55.4 *
Male newborns	54.2	60.1	52.4 *
Birthweight (grams), Mean (SD)	3207 (349)	3232 (365)	3200 (344)
Birthweight >3,600 g	10.5	12.9	9.8
Gestational age, Mean (SD), weeks	38.9 (0.9)	38.8 (0.9)	38.9 (0.9)
Newborns received skin-to-skin at birth	98.3	98.9	98.1
Skin-to-skin contact at least 90 min	92.4	89.6	93.3 *
Born to mothers with underlying medical condition	2.3	2.1	2.4
Born to mothers with breastfeeding difficulties during the hospital stay	16.0	11.2	17.4 *

Notes: Data were presented as % or mean (SD) when specified. Abbreviations: *n*: Sample number; DNHWC: Da Nang Hospital for Women and Children; PDM: Pasteurized donor milk; SD: Standard deviation; VND: Vietnam Dong, conversion rate of USD 1 for 22,900 VND. * Statistically different from newborns who used PDM, two-sided chi-squared tests for the prevalence and two-sided *t*-test for the mean.

**Table 3 nutrients-13-01151-t003:** Factors (adjusted OR; 95% CI) associated with the use and prolonged use of pasteurized donor milk among healthy newborns in postnatal wards.

	Used PDM (*n* = 2440)	Prolonged Use of PDM (*n* = 566)
Mothers:		
Age (year)	1.00 (0.98, 1.03)	1.00 (0.95, 1.06)
Farmer or worker (vs. other jobs)	0.71 * (0.54, 0.93)	0.25 * (0.08, 0.72)
Secondary school education or less (vs. college or higher education)	0.74 (0.48, 1.13)	0.85 (0.21, 3.35)
Family income <10 million VND (vs. ≥10 million VND)	0.67 *** (0.55, 0.82)	0.68 (0.39, 1.19)
First-time mother (vs. had a child prior)	1.09 (0.88, 1.36)	3.39 *** (1.92, 6.01)
Prenatal visit at DNHWC (vs. at other hospitals)	1.28 (1.04, 1.57)	1.37 (0.75, 2.5)
Cesarean birth (vs. vaginal birth)	2.05 *** (1.66, 2.55)	2.09 * (1.02, 4.28)
Newborns:		
Male gender (vs. female)	1.33 ** (1.09, 1.62)	1.35 (0.76, 2.37)
Birthweight >3600 g (vs. ≤3600 g)	1.20 (0.89, 1.63)	3.02 ** (1.55, 5.89)
Gestational age (weeks)	0.93 (0.83, 1.04)	0.75 (0.56, 1.02)
Duration of skin-to-skin ≥90 min (vs. <90 min)	0.54 *** (0.39, 0.76)	0.52 (0.18, 1.54)

Abbreviations: *n*: Sample number; DNHWC: Da Nang Hospital for Women and Children; PDM: Pasteurized donor milk; SD: Standard deviation; VND: Vietnam Dong, conversion rate of USD 1 for 22,900 VND. Among 2440 newborns, 566 used PDM, and 70 used PDM for more than 48 h (prolonged use). Adjusted odds ratios (OR) and 95% CI from multiple logistic regression models, controlled for covariates in this table. Significantly different from the null value (OR of 1): * *p* < 0.05, ** *p* < 0.01, *** *p* < 0.001.

## Data Availability

Requests for data may be directed to the corresponding author and are subject to institutional data use agreements.

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
