# Peer review of "Factors Associated with the Use of Pasteurized Donor Milk for Healthy Newborns: Experience from the First Human Milk Bank in Vietnam"

_nutrients, 2021, doi:10.3390/nu13041151_

Round 1
Reviewer 1 Report
The manuscript describes the benefits of prescribing donor milk to healthy newborns after vaginal and cesarean births on breast-feeding at hospital discharge in Vietnam. The concept is of interest and seems to prove beneficial when comparing to the state in 2016. The study lacks mechanistic insight as no physiologic or health parameters were tested (e.g. oxytocin). However, this was not the scope of the current manuscript but would be valuable for future studies and research.
Title:
- I suggest replacing "characteristics" with a term like "metadata" since it was more the mothers that were characterized (job status etc.).
M&M:
- L136: Access instead of Assess
- L162/163: provide names of R packages and versions
Results:
- L184: 1867 newborns; probably the authors have interviewed the mothers
- L192: add brackets (Table 1)
- Table 2: it seems that the majority of newborns have been male. Please comment on that where appropriate in M&M or discussion section. The ratio of female to male newborns in the entire data set was approximately 50:50?
Discussion:
- L228: Please comment on private payments to the general hospital stay.
- L270/271: PDM has to be paid privately. Please state this important issue prominently in Abstract and Conclusion sections.
Author Response
The manuscript describes the benefits of prescribing donor milk to healthy newborns after vaginal and cesarean births on breast-feeding at hospital discharge in Vietnam. The concept is of interest and seems to prove beneficial when comparing to the state in 2016. The study lacks mechanistic insight as no physiologic or health parameters were tested (e.g. oxytocin). However, this was not the scope of the current manuscript but would be valuable for future studies and research.
Response: Thank you very much.
Title:
- I suggest replacing "characteristics" with a term like "metadata" since it was more the mothers that were characterized (job status etc.).
Response:
With the changes in the scope of the manuscript based on the comment from Reviewer 2, we changed the title to “Factors associated with the use of pasteurized donor milk for healthy newborns: experience from the first human milk bank in Vietnam.”
M&M:
- L136: Access instead of Assess
Response: We revised as suggested.
- L162/163: provide names of R packages and versions
Response: We added as suggested.
Results:
- L184: 1867 newborns; probably the authors have interviewed the mothers
Response: We changed to mothers.
- L192: add brackets (Table 1)
Response: We revised as suggested.
- Table 2: it seems that the majority of newborns have been male. Please comment on that where appropriate in M&M or discussion section. The ratio of female to male newborns in the entire data set was approximately 50:50?
Response:
- Based on the monitoring data in the first 4 years of operation at the HMB at DNHWC 16,235 newborns received PDM. Of 11,162 newborns receiving PDM in postnatal wards, 54.4% were male newborns, which showed that male newborns were more likely to receive PDM. We did not comment on these data given it is not the main focus of this study. Further studies/ analyses will be needed.
- The prevalence of male newborns receiving PDM in postnatal wards from our sample is 60% which is higher than the overall prevalence (54.4%), which indicates that the sample might not be representative for all newborns receiving PDM at the hospital. Representativeness of the sample was mentioned in study limitations.
Discussion:
- L228: Please comment on private payments to the general hospital stay.
Response: We included the statement in the discussion. All medical expenses for children aged 6 years or less will be covered by health insurance. We made it clearer in the text (lines 327-328).
- L270/271: PDM has to be paid privately. Please state this important issue prominently in Abstract and Conclusion sections.
Response: We included in the abstract and conclusions as suggested (lines 21, 35, 91 and 375). We also discussed about the payment in discussion (lines 325-332).

Reviewer 2 Report
Nutrients_1128783
Tran et al present a study that evaluated the use of pasteurized human milk (PDM) for healthy term born infants born at a tertiary hospital in Da Nang, Vietnam. They conclude that prolonged use of PDM was associated with cesarean section and maternal socio-economic status. Authors suggest that reduction in caesarean section rates and communication about human milk bank guidelines as well as improvement of breastfeeding support is needed.
While the data presented are not completely new and in line with other publications, the results could be very valuable for intern use of the hospital to audit their own breastfeeding support guideline, as well as the international discussion to use PDM for healthy term infants. However to be acceptable for publication the authors should provide essentially more details concerning the practice of breastfeeding support, concerning the group of infant/mother pairs that did not use PDM and the Discussion section should be more focused to fit the final conclusion.
Major remarks
- in general:
While it is obvious that human milk is the best for every newborn infant, it is also known that PDM is only ‘second’ best as a result of pasteurization. Therefore the use is only acceptable for high-risk VLBW infants for prevention of necrotising enterocolitis. This is not the case for healthy newborns and furthermore pasteurization decreases the immunologic activity and therefore misses the valuable protection of ‘mothers own milk’ (MOM). The aim of exclusive breastfeeding should not be to provide 100% human milk but maximize the use of MOM. Offering PDM to healthy infants causes a negative incentive for all new mothers. This should be the clear message of the data presented. However while authors probably would agree to this statement (and one can read it in between the lines) they also state that all mother’s who used PDM for their infants left the hospital with exclusive breastfeeding and thus suggested that the current use of PDM is permissible. Exclusive breastfeeding actually cannot be validated by the data presented in this manuscript. How did authors control that after discharge from hospital exclusive breastfeeding was continued and for how long?
- research population
For this study authors interviewed 2440 mothers of whom 566 mothers used PDM. Authors only present data of the latter 566 women/infant combination. Although authors state that data of the remaining participants will be presented in a following manuscript, the current data are impossible to interpret without a control group of mothers who did not use PDM. Therefore it is unacceptable to not present equivalent data on the remaining 1874 mothers.
In detail:
Abstract:
Line 20: analysis study -> prospective observational study
Line 31 – 37: try to focus the conclusion
Line 46 - 48: author mention breastfeeding ‘in the first hour of life’ and that separation is an important reason for lack of success in breastfeeding. Actually breastfeeding should occur from the first hour onwards, separation of mother and child should be minimized, the infant should be breastfed on demand or at least every 2 hours and if there is doubt that the infant sucked sufficiently the mother should be helped in use of a breast pump, while her child can get the pumped milk. All efforts should be made to support and increase the self confidence of new mothers that they are functioning adequate.
Line 60 – 65: all references refer to the use of PDM in VLBW infants which is not the case in the current study
Line 67: PDM ‘can’ be prescribed: suggestion: alter ‘can’ to ‘is’ because this practise is actually questionable, adequate support of mothers probably would be the better option.
Question concerning the situation in Da Nang: Who is running the milk bank? Do donor mothers receive money for their donated milk? Is there an economic incentive to distribute PDM to healthy infants? This should be described either in the introduction or method section.
Line 91: authors might add: the study findings can be used to further improve breastfeeding strategies.
Method:
As described above, it is not clear whether infants/mothers got the opportunity to breastfeed on demand, at least every second hour and/or whether mothers were offered breast pumps. This should be mentioned in detail.
Why was PDM offered as early as 6 hours after birth? It may be seen as perfectly normal that lactation has not started at that moment and healthy infants have enough reserves to overcome several hours without nutrition, assuming that none of the infants did have a medical reason to offer nutritional intake, such as hypoglycaemia or dehydration. Not offering feeding even may stimulate the sucking. As mentioned above offering PDM that early undermines the self-confidence of mothers. The indication for starting PDM should be explained more in detail, if possible mentioned in the results. However, regarding the fact that a great number of low-income mothers did not use PDM without leading to medical deterioration of their child, the use of PDM probably may be interpreted as being medically unnecessary. This also should be stated more clearly in the discussion section.
Results:
How can authors prove 97% exclusively breastfeeding. Was there (and how long was) an interval between last use of PDM and fully exclusive breastfeeding? Did authors ask for mode of feeding after discharge? This should be mentioned. What was the duration of hospital stay per group?
Discussion
The discussion is quite lengthy and can be reduced to half of the words. It is recommended to consider to focus on aspects that highlight unnecessary and undesirable use of PDM.
Cesaerean section and pain should not be reason for delay in breastfeeding, support of mothers might be changed, first-time mothers probably need more support, and the counselling of mothers with high job status should support their self confidence in that they are just doing fine and their child is just doing well.
Author Response
Reviewer 2
Tran et al present a study that evaluated the use of pasteurized human milk (PDM) for healthy term born infants born at a tertiary hospital in Da Nang, Vietnam. They conclude that prolonged use of PDM was associated with cesarean section and maternal socio-economic status. Authors suggest that reduction in caesarean section rates and communication about human milk bank guidelines as well as improvement of breastfeeding support is needed.
Response: Thank you.
While the data presented are not completely new and in line with other publications, the results could be very valuable for intern use of the hospital to audit their own breastfeeding support guideline, as well as the international discussion to use PDM for healthy term infants. However, to be acceptable for publication the authors should provide essentially more details concerning the practice of breastfeeding support, concerning the group of infant/mother pairs that did not use PDM and the Discussion section should be more focused to fit the final conclusion.
Response:
- We revised to include suggested information (Tables 2, 3, and corresponding text (lines 215-222; 234-242)).
- We also shortened the discussion and made it more focused as suggested with the discussion including reasons for non-medical use of PDM and steps required to address problems (lines 228-332).
Major remarks
- in general:
While it is obvious that human milk is the best for every newborn infant, it is also known that PDM is only ‘second’ best as a result of pasteurization. Therefore the use is only acceptable for high-risk VLBW infants for prevention of necrotising enterocolitis. This is not the case for healthy newborns and furthermore pasteurization decreases the immunologic activity and therefore misses the valuable protection of ‘mothers own milk’ (MOM). The aim of exclusive breastfeeding should not be to provide 100% human milk but maximize the use of MOM. Offering PDM to healthy infants causes a negative incentive for all new mothers. This should be the clear message of the data presented. However while authors probably would agree to this statement (and one can read it in between the lines) they also state that all mother’s who used PDM for their infants left the hospital with exclusive breastfeeding and thus suggested that the current use of PDM is permissible. Exclusive breastfeeding actually cannot be validated by the data presented in this manuscript. How did authors control that after discharge from hospital exclusive breastfeeding was continued and for how long?
Response:
- We agree about the importance of mothers’ own milk and need to promote breastfeeding. The purpose of our study is to raise the red flag about the use of PDM, which can undermine mothers’ own milk supply and potentially limit breastfeeding. We made it clearer in the text (Lines 18-20, 92-94, 228-332).
- Nonetheless, we believe that PDM is a better option, when necessary, than infant formula, all agree that PDM better than infant formula (lines 333-346).
- We agree that the questionnaire is not standardized to evaluate exclusive breastfeeding at discharge (e.g., in the previous 24 hours). Thus, we excluded the statement about exclusive breastfeeding at or after discharge.
- research population
For this study authors interviewed 2440 mothers of whom 566 mothers used PDM. Authors only present data of the latter 566 women/infant combination. Although authors state that data of the remaining participants will be presented in a following manuscript, the current data are impossible to interpret without a control group of mothers who did not use PDM. Therefore, it is unacceptable to not present equivalent data on the remaining 1874 mothers.
Response: We revised the manuscript to include the complete sample (n=2,440) and include a new aim on associate factors of using PDM (Tables 2, 3, and corresponding text (lines 215-222; 234-242).
In detail:
Abstract:
Line 20: analysis study -> prospective observational study
Response: We revised as suggested.
Line 31 – 37: try to focus the conclusion
Response: We revised to make the conclusion more focused (lines 34-38).
Line 46 - 48: author mention breastfeeding ‘in the first hour of life’ and that separation is an important reason for lack of success in breastfeeding. Actually breastfeeding should occur from the first hour onwards, separation of mother and child should be minimized, the infant should be breastfed on demand or at least every 2 hours and if there is doubt that the infant sucked sufficiently the mother should be helped in use of a breast pump, while her child can get the pumped milk. All efforts should be made to support and increase the self confidence of new mothers that they are functioning adequate.
Response: We agree with the reviewer about the importance of early initiation of breastfeeding. We presented in the discussion (lines 271-275 and 281-202). For the purpose of the introduction, this paragraph is to indicate about the need of PDM for select newborns: mothers’ own milk not available; and other factors mentioned are those that might contribute to mothers’ own milk not available.
Line 60 – 65: all references refer to the use of PDM in VLBW infants which is not the case in the current study
Response: For the purpose of introduction, we started with the current recommendations from WHO, UNICEF about the use of PDM for VLBW. In the following paragraph, we wrote about the use of PDM for healthy newborns.
Line 67: PDM ‘can’ be prescribed: suggestion: alter ‘can’ to ‘is’ because this practise is actually questionable, adequate support of mothers probably would be the better option.
Response: Revised to “is” as suggested.
Question concerning the situation in Da Nang: Who is running the milk bank? Do donor mothers receive money for their donated milk? Is there an economic incentive to distribute PDM to healthy infants? This should be described either in the introduction or method section.
Response: We provided suggested information in the methods (lines 115-124) with a citation to our previous published paper.
Line 91: authors might add: the study findings can be used to further improve breastfeeding strategies.
Response: We added the statement as suggested (lines 92-94). Also, for your information, the purpose of our manuscript is to flag the unnecessary use of PDM. We, however, think that it is also needed to have further studies and international guidelines relating to this topic (lines 347-349).
Method:
As described above, it is not clear whether infants/mothers got the opportunity to breastfeed on demand, at least every second hour and/or whether mothers were offered breast pumps. This should be mentioned in detail.
Response: We revised to add the information (lines 110-114)
Why was PDM offered as early as 6 hours after birth? It may be seen as perfectly normal that lactation has not started at that moment and healthy infants have enough reserves to overcome several hours without nutrition, assuming that none of the infants did have a medical reason to offer nutritional intake, such as hypoglycaemia or dehydration. Not offering feeding even may stimulate the sucking. As mentioned above offering PDM that early undermines the self-confidence of mothers. The indication for starting PDM should be explained more in detail, if possible mentioned in the results. However, regarding the fact that a great number of low-income mothers did not use PDM without leading to medical deterioration of their child, the use of PDM probably may be interpreted as being medically unnecessary. This also should be stated more clearly in the discussion section.
Response: We agree with the reviewer that giving PDM early and to a large number of healthy newborns is the practice that need to be controlled for. We made it clearer in the discussion (lines 258-265).
Results:
How can authors prove 97% exclusively breastfeeding. Was there (and how long was) an interval between last use of PDM and fully exclusive breastfeeding? Did authors ask for mode of feeding after discharge? This should be mentioned. What was the duration of hospital stay per group?
Response: We excluded all statements / assumptions about the impact of using PDM because they were not supported by our study design. We did not use standardized a 24-hour recall questionnaire (e.g., from MICS) to evaluate breastfeeding at discharge. We did not follow up with the mothers about breastfeeding practices after discharge.
Discussion
The discussion is quite lengthy and can be reduced to half of the words. It is recommended to consider to focus on aspects that highlight unnecessary and undesirable use of PDM.
Response: We revised to shorten the Discussion and focused unnecessary and undesirable use of PDM. However, we indicated the gaps in knowledge relating to the use of PDM for healthy newborns and the need for relevant Guidelines.
Cesaerean section and pain should not be reason for delay in breastfeeding, support of mothers might be changed, first-time mothers probably need more support, and the counselling of mothers with high job status should support their self confidence in that they are just doing fine and their child is just doing well.
Response: We shortened this paragraph, and made it clear that these factors should not be a reason to not breastfeed included it after the discussion about unnecessary and undesirable use of PDM (lines 27

Round 2
Reviewer 2 Report
comments to the authors are presented in the attachment
